# Does voluntary practice improve the outcome of an OSCE in undergraduate medical studies? A Propensity Score Matching approach

**Stefanos A. Tsikas**[1]*, **Kambiz Afshar**[2], **Volkhard Fischer**[1]

1 Dean of Studies Office – Academic Controlling, Hannover Medical School, Hannover, Lower Saxony, Germany, 2 Institute for General Practice and Palliative Care, Hannover Medical School, Hannover, Lower Saxony, Germany

* Tsikas.Stefanos@mh-hannover.de

**Data Availability Statement:** All relevant data are within the manuscript and its Supporting information files.

## Abstract

In Objective Structured Clinical Examinations (OSCE), students have to demonstrate proficiency in a wide array of medical knowledge and different skills, ranging from physical examinations to medical-practical skills and doctor-patient interactions. In this study, we empirically test the concept 'assessment drives learning' and investigate whether an OSCE motivates extracurricular, voluntary free practice (FP) of specific skills in a Skills Lab, and whether this has positive treatment effects on exam success in the respective parts of the OSCE. To explore causal inference with observational data, we used Propensity Score Matching (PSM) to generate a control and a treatment group that only differed in their practice behavior. For internal examinations and practical skills such as venous catheter placement and IM injections, we find strong, positive effects of FP that can result in a grade-jump. We further show that the presence and strength of effects depends on the complexity and type of the task. For instance, we find no effect for practicing venipuncture, and performance in communicative skills is associated with the willingness to repeatedly engage with instructional contents inside and outside the Skills Lab, and not with targeted practice of specific skills. We conclude that the anticipation of the complex OSCE is effective in motivating students to engage with a wide range of competencies crucial to the medical profession, and that this engagement has positive effects on exam success. However, consistent practice throughout the study program is necessary to sustain and nurture the acquired skills.

## Introduction

Acquiring basic medical knowledge and learning procedural as well as technical skills constitute the prerequisite for the development of professional competencies during medical studies [1]. In particular, these principles comprise of physical examination techniques (e.g. examination of the lung, heart, or abdomen), skills for conducting a conversation with patients (e.g. taking a medical history or conveying a diagnosis), as well as practical skills for routine medical activities (e.g. venipuncture for blood taking, ECG interpretation, intramuscular injection). It

**Funding:** The author(s) received no specific funding for this work.

**Competing interests:** The authors have declared that no competing interests exist.

is beneficial if the knowledge and skills are taught at an early phase, so that they can be trained sustainably during medical studies, especially in the clinical parts, and consolidated until graduation [2–5].

For the majority of medical students it is self-evident that these skills are an essential part of physicians' every day work. Consequently, a high intrinsic motivation to learn and to adopt should be assumed. Nevertheless, it is also necessary to examine the quality of acquired skills before allowing medical students to apply the learned contents in direct encounters with patients.

This link between learning and assessments has been formalized in the concept 'assessment drives learning' (or 'assessment for learning') [2, 6–8]. This framework suggests that the topics and design of exams, for example summative assessments, play an important role in terms of students' learning behavior, as they motivate and promote exam preparation and engagement with learning contents, especially if the examination is accompanied by a grade [9, 10]. In terms of constructive alignment, the assessment procedure should be selected to match curricular learning objectives and didactic teaching methods [11]. However, it is still unclear how exactly the examination procedure affects learning activities and how students proceed to prepare for an examination. At the Hannover Medical School (MHH, the site of our study), for instance, the overall curriculum was not explicitly orientated towards the objectives of the final state examination in the sense of Kane [12], but was rather established in an evolutionary way [13].

An adequate assessment procedure that meets the above prerequisites and that can be used to investigate the links between exam procedure, preparation and exam success empirically is an objective structured clinical examination (OSCE) [14] with simulated patients and simulators [15, 16]. In this assessment procedure, medical students have to demonstrate the learned content, according to stage three of Miller's pyramid ("show how" [17]). As the OSCE is a practical assessment procedure where students have to attest that they know how to translate their competencies into action, medical students have to prepare themselves accordingly. According to 'assessment drives learning', the choice of the assessment procedure influences medical students' learning behavior [2, 6–8]. In addition to teaching courses, where contents are presented usually for the first time, medical students have to practice these contents on further occasions to become proficient practitioners. At many medical faculties, students have the opportunity to train in voluntary practice sessions on their own or with fellow students in so-called Skills labs.

In this paper, our objective is to empirically test the concept of 'assessment drives learning' in context of an OSCE and a skills lab that is used for exam preparation and learning of practical-medical skills.

While a general relationship between training practical skills and the confidence and quality of their application seems intuitive, it is also ambiguous: on the one hand, it is reasonable to assume that diligent students not only perform better in exams but also take advantage of practice opportunities more frequently and reliably. On the other hand, specifically in the context of an OSCE, existing experience in medical professions may lead to voluntary, additional practice not being necessary or less frequent to achieve a good exam outcome.

Thus, causal links between voluntary learning, competence acquisition ('knows how'), and performance in examinations ('shows how') are challenging to identify and still largely unexplored. In our study at MHH, we aim to enable the possibility to draw causal inferences by mitigating the described biases and confounding factors. We do so by using Propensity Score Matching (PSM) to create socio-demographically similar groups that differ only in whether a specific aspect of medical knowledge or technical skills was practiced in the Skills lab, and then compare the OSCE results of these groups.

## Materials and methods

### The MHH OSCE

The OSCE at MHH takes place at the end of the second year of study and serves as the examination for the 'Diagnostic Methods' module. Designed to encompass a broad spectrum of practical medical activities, the OSCE gives students a formative and summative feedback on their current performance level in the study program [18].

The OSCE primarily utilizes standardized patients, and medical manikins for the 'Medical Skills' component. The examination takes place in the MHH Skills lab.

The 'Physical examination' part of the OSCE comprises three internal medicine assessments, selecting from five possible areas (see Fig 1). The 'Neurological examination' involves a check-up of the nervous system. The 'Communication' component consists of two stations for medical consultations with standardized patients: 1) gathering patients' medical history and 2) delivering bad news and diagnosis of a severe illness [19]. In the 'Medical Skills' category, students demonstrate practical skills in two out of six areas (see Fig 1). An additional 25 points are allocated for the structured diagnosis of an X-ray [20]. As this skill is not trained in the Skills lab, it is not part of this study.

Grading is based on achieving a minimum of 90% for the best grade (1, 'A'), 80% for the grade 2 ('B'), and 70% for a 3 ('C'). For the OSCE outcome measure, we use the percentage of points achieved relative to the maximum possible score overall and at individual stations.

Lecturers of the module Diagnostic Methods usually introduce the Skills lab and the opportunity for free practice (FP), and emphasize its usefulness to prepare adequately for the OSCE. Time slots (45 minutes) and subjects are booked online.

### Questionnaire

The core part of our questionnaire on FP-behavior, detailed in Supplement 4 in S1 File, was the self-assessment if and how often the Skills lab was used for FP, and which OSCE stations

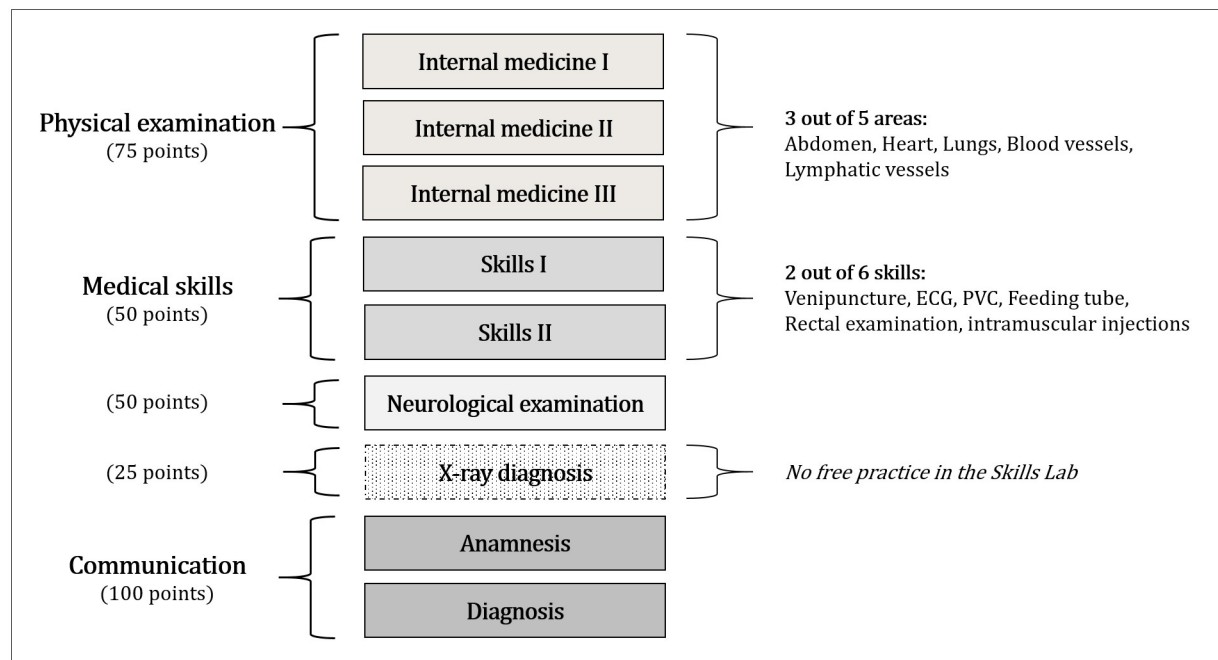

**Fig 1. The OSCE at MHH.**

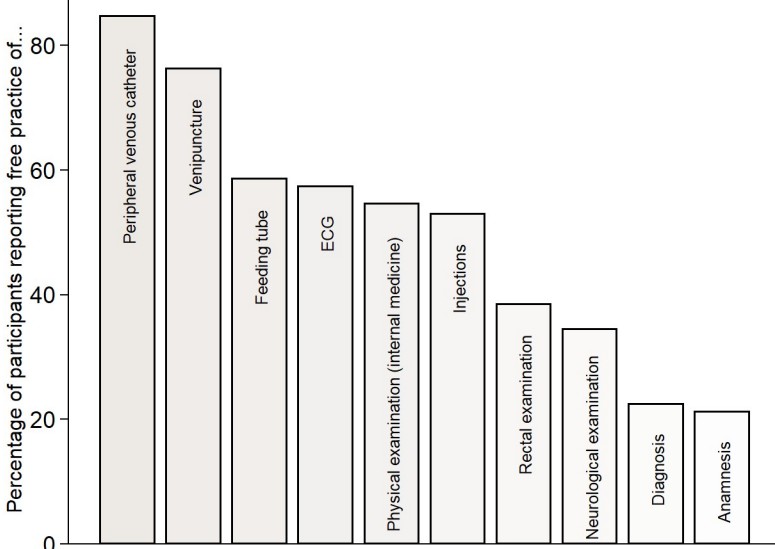

**Fig 2. Share of participants who engaged in free practice (FP) in the depicted skills/topics.** Answering "yes" means that the respective medical skill or OSCE topic was practiced at least once. On the questionnaire, the number of times a skill was trained was not polled. In 2023, it was asked whether physical and neurological examinations as well as anamnesis and diagnosis were practiced in another extracurricular context than the Skills Lab. The response "yes" was more common than to the question narrowed to the Skills Lab. In Fig 1, we combined FP in- and outside the Skills Lab for these OSCE topics.

and/or medical skills were trained. Fig 2 in the results section provides an overview of the skills and OSCE topics that respondents practiced, as indicated through a multiple-choice item in the questionnaire. The questionnaire did not inquire about the frequency of training for each specific skill. In our analysis, we categorized a skill as 'not trained' if the corresponding checkbox in the questionnaire was left unselected or if respondents did not use FP at all. In 2023, we additionally asked about FP of communicative skills and neurological as well as physical examinations outside the Skills lab. In Fig 2 and for our empirical analyses, we pooled FP in- and outside the Skills lab, but conducted robustness checks where we excluded students who only reported FP in private surroundings in 2023.

Other important information acquired with the questionnaire were the status of vocational training, prior experiences with FP, and the motivations to visit the Skills lab. The completion of the questionnaire took no longer than five minutes on average.

## Data

We sent the online-questionnaire to all students registered for participation in the OSCE in July 2021, 2022 and 2023, shortly before the commencement of the examination. The polling remained accessible until approximately one week after the conclusion of the OSCE. In advance, students were thoroughly briefed via Email on the study's procedures, objectives, and the handling of data, including aspects of data privacy. Within the questionnaire, an informed consent clause was incorporated, requiring participants to agree to the merger of responses and their exam results. The informed consent and study regulations governing the use of personal data for evaluation/research and quality assurance purposes (in particular sect. 14, para. 1–5 '*MHH Immatrikulationsordnung*' and sect. 17, para. 3 NHG (Higher Education Act in Lower Saxony, Germany)) rendered a separate approval by an ethics committee unnecessary.

Information on gender, age, nationality and GPA were retrieved from official student statistics. Detailed OSCE results were provided by the managers of the module 'Diagnostic Methods'. After data collection, a fully anonymized dataset with the consenting participants was generated for data analysis. The research presented here is in accordance with the Helsinki Declaration.

Overall, 306 participants responded to the questionnaire, constituting approximately 32% of all students reached via email. Among the respondents, 82%, or 251 individuals, granted consent for the utilization of their data in the study. Six individuals from this pool did not partake in the OSCE, for reasons such as illness. Thus, the effective sample size amounted to N = 245.

In our sample, 69% of the respondents were female, the average age at enrollment was 22 years (SD: 3.95). The school-leaving grade, the primary criterion for access to medical school, measured on a scale from 100 (best, 'A') to 400 (worst passing grade, 'D'), averaged at 156 (SD: 55). Overall, 40% of participants indicated having commenced vocational training.

Regarding sociodemographic characteristics, we find a substantial degree of representativeness compared to the entire OSCE cohorts spanning 2021–2023 (see Supplement 1 in S1 File). Turning to OSCE results, our study participants performed significantly better across various segments of the clinical examination in comparison to the average results of all students from 2021 to 2023. However, the observed differences, typically ranging between 1–2 percentage points, are relatively modest.

## Propensity Score Matching

Propensity Score Matching (PSM) aims to identify, within a (large) group of non-participants, individuals who are similar to a group of participants in relevant observed characteristics that influence both the propensity to receive (or select into) a treatment and the outcome itself [21–26].

Propensity scores are a unidimensional balancing score defined as the probability of participation (in a treatment or an intervention) conditional on a set of observed characteristics [27]. Propensity Scores are non-parametrically estimated with a logistic regression model. The selection of our matching characteristics is explained and justified in the next subsection. Considering the relatively modest sample size in our study, especially for some medical skills (see Fig 2), and the limited number of covariates available to estimate propensity scores, we employ PSM with the following considerations:

First, we use a non-parametric kernel estimator as our matching algorithm, which calculates weighted averages of all individuals in the control group [28, 29]. The kernel function assigns higher weights to pairs with smaller distances, indicating more similarity, and lower weights for less similarity. Kernel matching allows for more matches, resulting in reduced variance and precise estimates, but may yield poorer (or biased) counterfactuals, because also subjects with slightly different propensity scores can be matched [27, 29]. Second, we allow for matching with replacement, where an individual in the control group can be matched with a treated subject more than once. This approach reduces the requirement for the number of distinct non-participants–which is beneficial for small sample sizes–but increases the variance of the estimator [28]. Thirdly, we enforce a strict rule of common support, ensuring that subjects with similar characteristics have a positive probability of selecting either the control or the treatment group, which is of particular importance for Kernel matching [30]. Thus, we exclude observations in the treatment group whose propensity score is outside the range of scores in the control group, i.e. less than the minimum or higher than the maximum propensity score. While this may reduce the sample size in some instances, it enables meaningful comparisons

between the treatment and control groups and ensures accurate estimation of treatment effects, as causal effects are only defined within the region of common support [29].

Given our study design, where participants self-select into either control or treatment group, we focus on measuring the Average Treatment Effect on the Treated (ATT). The ATT provides causal insights into the effectiveness of training in the Skills Lab for those who actually utilized it, compared to similar students who decided not to practice [26, 31–33].

## Choice of matching parameters

The covariates observed in our study can be broadly categorized into two main groups: academic achievements prior to the OSCE and sociodemographic characteristics. It is well documented in the literature that GPAs have strong predictive validity for study success in medical school and higher education in general [34–36]. Exam outcomes at university also serve as predictors of future exam performance, including OSCEs [18, 37, 38]. Study success could also be correlated to voluntary practice: while diligent students may be motivated to put in extra effort to achieve the best possible results, struggling students may see free practice as an opportunity to gain confidence. For these reasons, we incorporate high school GPAs and exam results from the first two years of study (including both written and oral exams) into our estimation of propensity scores. Tsikas & Fischer [38] have demonstrated that a timely study progress in the first years of study has sound predictive validity for later study success. To accommodate this finding in our study, we additionally include a binary indicator for matching that takes the value of 1 if all exams scheduled before the OSCE had been passed, and 0 if one or several exams had been failed or not yet taken.

Another parameter we include is vocational training: students with professional experience in healthcare are less likely to practice certain skills assessed in the OSCE. However, due to their existing experience, this does not necessarily have a negative impact on OSCE performance, although these students generally have lower GPAs and consistently perform less well in multiple-choice tests [38–41].

Approximately two-thirds of the student population consists of females, who are more likely to respond to questionnaires and more likely to utilize the Skills lab for FP (see Supplement 2 in S1 File). Although differences in study success between men and women are often negligible, we match groups based on their gender.

## Results

### Descriptive statistics

In total, 89% of respondents reported visiting the Skills Lab for Free Practice (FP) at least once before the OSCE, with an average of 3.22 visits (SD: 2.68). The primary motivation behind utilizing the Skills lab was preparation for the OSCE, as highlighted by 90% of respondents who emphasized the significant role the exam played in their decision for FP.

Over 90% of participants in 2023 stated practicing physical examinations outside the university setting, the numbers for neurological examinations are 78% and for communicative skills 60%, respectively. Notably, the numbers for communicative skills were markedly higher outside the Skills lab compared to voluntary practice within its confines.

In Fig 2, it becomes evident that skills requiring the Skills lab were practiced more frequently, with particular emphasis on PVC and venipuncture. Conversely, neurological examinations and communicative skills, which collectively contribute to half of the OSCE points (see Fig 1), were not practiced as frequently.

In terms of medical skills, our analysis reveals a correlation between the rate of FP and the frequency of examination in the OSCE. For instance, 55% of students were examined in the

placement of a PVC, the ratio for venipuncture and (intramuscular) injections is 35%. A fourth of study participants was examined in the insertion of a feeding tube, 18–19% of participants in the skills ECG and rectal examination.

Female participants exhibited a notably higher frequency of visits to the Skills lab compared to their male counterparts, a trend observed both overall and when considering the specific skills outlined in Fig 1. Conversely, students with vocational training or prior work experience tended to utilize the Skills lab less frequently, with the most pronounced differences observed in medical skills such as venipuncture, PVC, and injections (see Table II-1a in Supplement 2 in S1 File). There are low to moderate negative correlations between poorer school leaving and exam grades and FP (Table II-1b, Supplement 2 in S1 File).

In analyzing OSCE results, no significant differences were found based on participants' gender, vocational training status, or age, except in the context of physical examination (Tables II-2, Supplement 2 in S1 File). Predictive validity analyses revealed that a higher (thus, worse) school-leaving grade was indicative of less success in the overall OSCE and specifically in the components of physical and neurological examination. The school-leaving grade showed no correlation with overall performance in medical skills, and only low correlations were found for ECG and rectal examination.

Success in the written and oral exams preceding the clinical examination demonstrated strong predictive validity for OSCE outcomes (Table II-2c, Supplement 2 in S1 File), encompassing both completion rate and average grade achieved. Weaker correlations were noted for medical skills ($r = -0.155$; $p = 0.015$) and communicative skills ($r = -0.189$; $p = 0.003$).

In Table 1, we address the question of the relative importance of using the Skills lab for FP in general, including the number of visits, versus targeted training on the specific topics examined in the OSCE. As a potential proxy for diligence, commitment to exams, and overall success in the study program, the former demonstrates moderate positive correlations with OSCE success. Specifically, there is a correlation of $r = 0.310$ ($p<0.001$) between the percentage achieved in the OSCE and engagement in FP, and a correlation of $r = 0.371$ ($p<0.001$) with the number of visits to the Skills lab.

Students with poorer exam results preceding the OSCE were less likely to have visited the Skills lab, a trend potentially influenced by the professionally experienced, a consideration we

**Table 1. Associations between Free Practice (FP) measures and OSCE results (ANOVA).**

|  | OSCE (N = 245) | Physical exam. (N = 245) | Neurological exam. (N = 245) | Anamnesis (N = 245) | Diagnosis (N = 245) |  |
|---|---|---|---|---|---|---|
| FP | 24.91 (0.000) | 3.88 (0.050) | 0.77 (0.381) | 8.84 (0.003) | 0.22 (0.640) |  |
| FP (spec. Skill) |  | 23.32 (0.000) | 0.59 (0.445) | 0.03 (0.873) | 0.00 (0.994) |  |
| FP (no. of visits) | 2.85 (0.192) | 1.71 (0.192) | 7.67 (0.006) | 1.62 (0.205) | 6.42 (0.012) |  |
| Adj. $R^2$ | 0.157 | 0.151 | 0.045 | 0.053 | 0.025 |  |
|  | Venipuncture (N = 86) | ECG (N = 48) | PVC (N = 135) | Feeding tube (N = 61) | Rectal exam. (N = 45) | Injections (N = 84) |
| FP | 0.39 (0.533) | 2.53 (0.119) | 4.40 (0.038) | 0.67 (0.416) | 1.54 (0.221) | 1.31 (0.256) |
| FP (spec. Skill) | 1.64 (0.204) | 0.00 (0.999) | 10.86 (0.001) | 0.35 (0.556) | 4.12 (0.049) | 12.86 (0.001) |
| FP (no. of visits) | 4.59 (0.035) | 1.07 (0.306) | 0.66 (0.418) | 1.61 (0.210) | 1.38 (0.247) | 0.21 (0.646) |
| Adj. $R^2$ | 0.027 | -0.002 | 0.092 | 0.039 | 0.132 | 0.113 |

ANOVA with the OSCE result overall and at the specific stations/medical skills as the dependent variable. The binary factors free practice (FP), the free practice of the specific station/skill (yes/no) and the number of visits to the Skills lab (coded as a continuous variable) are the predictors. P-values are in parentheses.

address in the Propensity Score Matching (PSM) analysis to account for potential confounders.

The ANOVA in Table 1 examine the strength of associations between OSCE results and three FP measures: 1) whether the Skills lab was used for FP at all, 2) the frequency of Skills lab utilization, and 3) whether FP occurred in the specific station or skill later examined in the OSCE. Table 1 reveals a robust and statistically significant association between OSCE outcomes and FP, although not with the number of visits to the Skills lab. For the physical examination and medical skills such as PVC, rectal examination, and injections, directed training of this specific skill emerges as more pivotal than the frequency of FP. Conversely, for other OSCE components, particularly anamnesis and diagnosis, the number of visits to the Skills lab is significantly correlated with the exam outcome, whereas specific training in communicative skills does not exhibit the same level of correlation.

## Treatment effects

Free Practice (FP) in Table 2 is the treatment group, consisting of participants who reported training for the specific OSCE station or medical skill, with the "No FP" participants serving as the control group. The varying number of observations for medical skills is due to each student being examined in two out of six different skills, and some medical skills being more frequently assessed than others. The 'unmatched' category represents the raw comparison between the groups that utilized FP and those that did not. The ATT, the 'average treatment effect on the treated' reflects the impact of FP after matching, i.e. the treatment effect.

In Table 2, a T-statistic > |1.96| corresponds to $p < 0.05$, indicating that the difference between the control and treatment groups is statistically significant. Matching may result in some subjects in the FP group not being paired with a counterpart in the No FP group due to propensity scores falling outside the range found in the control group (see Fig III-2 in Supplement 3 in S1 File). The discrepancy between utilized and total observations is denoted in the 'Common support' column in Table 2.

For both unmatched and matched group comparisons, FP is generally associated with a higher percentage achieved in the OSCE, with exceptions noted for venipuncture and ECG. Statistically significant differences are observed for Physical examination and the medical skills PVC, rectal examination, and IM injections.

Matching often reduces the T-statistic, signifying that included parameters influence and bias both the likelihood of using FP and the OSCE performance, although the impact size is at most moderate.

In the OSCE part 'Physical examination', there is a 5 percentage point difference, indicating that FP can elevate students from a grade 2 (equivalent to a 'B') to the best grade 1 ('A'). Performance in neurological examination and communication skills do not show a significant improvement with FP.

The placement of a peripheral venous catheter (PVC) appears to be the most challenging medical skill, with FP demonstrating a substantial ATT of almost ten percentage points in the OSCE. For rectal examinations and (intramuscular) injections, the control group achieves a strong 89% respectively 91%, but FP adds another 4.5% (6.3%). Statistically significant ATTs align with the fundamental findings from the ANOVA, corroborating the impact of FP on OSCE performance for specific skills and stations.

## Quality of matching & robustness checks

Propensity Score Matching (PSM) aims to create control and treatment groups based on selected variables that closely resemble each other when randomization is not feasible. Table 3

**Table 2. The effect of Free Practice (FP) on OSCE performance (PSM).**

| | FP | No FP | Difference | S.E. | T-Stat. | Common support |
|---|---|---|---|---|---|---|
| **Physical examination** | | | | | | |
| Unmatched | 90.85% | 85.02% | 5.83% | 0.965 | 6.05 | |
| ATT | 90.85% | 85.98% | 4.87% | 1.030 | 4.72 | 243/245 |
| **Neurological examination** | | | | | | |
| Unmatched | 85.56% | 83.36% | 2.20% | 1.509 | 1.46 | |
| ATT | 85.56% | 83.73% | 1.83% | 1.524 | 1.20 | 245/245 |
| **Anamnesis** | | | | | | |
| Unmatched | 83.96% | 83.38% | 0.58% | 1.554 | 0.38 | |
| ATT | 83.96% | 83.60% | 0.36% | 1.515 | 0.24 | 244/245 |
| **Diagnosis** | | | | | | |
| Unmatched | 84.71 | 84.37 | 0.34% | 1.217 | 0.28 | |
| ATT | 84.71 | 84.26 | 0.45% | 1.221 | 0.38 | 245/245 |
| **Medical Skills** | | | | | | |
| **Venipuncture** | | | | | | |
| Unmatched | 94.48% | 96.35% | -1.87% | 2.212 | -0.85 | |
| ATT | 93.76% | 96.53% | -2.77% | 2.542 | -1.09 | 57/86 |
| **ECG** | | | | | | |
| Unmatched | 92.46% | 92.91% | -0.45% | 2.872 | -0.16 | |
| ATT | 91.2% | 93.12% | -1.92% | 3.499 | -0.55 | 42/48 |
| **PVC** | | | | | | |
| Unmatched | 94.86% | 87.58% | 7.28% | 2.129 | 3.42 | |
| ATT | 94.53% | 84.55% | 9.98% | 4.966 | 2.01 | 109/135 |
| **Feeding tube** | | | | | | |
| Unmatched | 93.26% | 89.91% | 3.35% | 2.285 | 1.47 | |
| ATT | 92.00% | 89.28% | 2.72% | 2.893 | 0.94 | 52/61 |
| **Rectal examination** | | | | | | |
| Unmatched | 93.45% | 88.52% | 4.93% | 1.834 | 2.69 | |
| ATT | 93.85% | 89.30% | 4.55% | 2.281 | 1.99 | 36/45 |
| **Injections** | | | | | | |
| Unmatched | 97.06% | 91.77% | 5.29% | 1.562 | 3.39 | |
| ATT | 97.06% | 90.76% | 6.30% | 2.015 | 3.13 | 84/84 |

Average treatment effects on the treated (ATT) obtained from Propensity Score Matching with the parameters gender, vocational training, school-leaving grade, the average exam grade in the M1-phase and a binary indicator taking the value = 1 if all exams preceding the OSCE had been passed. Kernel matching (Epanechnikov kernel) with replacement and a strict condition of common support. For quality of matching tests, see Table 3 and Supplement 3 in S1 File. For propensity score estimates, see Supplement 2. A T-Statistic > |1.96| corresponds to $p < 0.05$. Common support denotes how many of the respective total respondents could be used for matching. "Unmatched" is a sample test for the groups Free Practice (FP) and No FP. S.E.: standard error.

illustrates how dissimilar or similar these FP and No FP groups are without and with matching. Successful matching should render the used covariates useless (statistically speaking) for predicting group membership. The Pseudo (Ps) $R^2$ (the coefficient of determination) should be close to 0, and the likelihood-ratio (LR) $\chi^2$ test should not be statistically significant.

Table 3 affirms the quality of matching in our study, because the explanatory power of the covariates is diminished to (almost) zero in most cases. Without it ('unmatched'), sociodemographic factors explain group membership relatively well, particularly concerning certain medical skills. As detailed in the descriptive analyses, this was largely influenced by the FP-

**Table 3. Quality of matching.**

|  |  | Ps R$^2$ | LR χ$^2$ | $p > \chi^2$ | Mean Bias | Median Bias |
|---|---|---|---|---|---|---|
| Physical examination | Unmatched | 0.029 | 9.66 | 0.086 | 19.0 | 22.7 |
|  | Matched | 0.000 | 0.05 | 1.000 | 0.9 | 0.7 |
| Neurological examination | Unmatched | 0.034 | 10.93 | 0.053 | 13.4 | 4.9 |
|  | Matched | 0.001 | 0.20 | 0.999 | 1.6 | 1.1 |
| Anamnesis | Unmatched | 0.008 | 2.11 | 0.833 | 7.3 | 5.6 |
|  | Matched | 0.002 | 0.25 | 0.998 | 3.1 | 2.1 |
| Diagnosis | Unmatched | 0.008 | 2.24 | 0.816 | 8.7 | 10.2 |
|  | Matched | 0.001 | 0.20 | 0.999 | 2.1 | 1.1 |
| Venipuncture | Unmatched | 0.158 | 15.76 | 0.008 | 48.2 | 62.6 |
|  | Matched | 0.017 | 1.63 | 0.898 | 13.0 | 11.6 |
| ECG | Unmatched | 0.074 | 4.89 | 0.429 | 25.3 | 18.9 |
|  | Matched | 0.010 | 0.58 | 0.989 | 6.1 | 6.8 |
| PVC | Unmatched | 0.202 | 22.18 | 0.000 | 53.2 | 61.2 |
|  | Matched | 0.049 | 12.19 | 0.032 | 14.7 | 7.8 |
| Feeding tube | Unmatched | 0.080 | 6.43 | 0.266 | 23.4 | 16.0 |
|  | Matched | 0.036 | 2.86 | 0.721 | 11.7 | 10.7 |
| Rectal examination | Unmatched | 0.145 | 9.04 | 0.108 | 25.2 | 13.5 |
|  | Matched | 0.027 | 0.96 | 0.966 | 14.9 | 14.7 |
| Injections | Unmatched | 0.058 | 6.57 | 0.254 | 29.3 | 33.9 |
|  | Matched | 0.001 | 0.08 | 1.000 | 2.2 | 2.3 |

Quality of matching tests post-estimation of propensity scores (see Supplement 2 in S1 File) and matching, see Table 2. Ps (Pseudo) R$^2$: coefficient of determination, i.e.: how well do the parameters used for matching predict the selection into the 'FP' and 'No FP' group. LR: Likelihood Ratio. Mean and Median Bias indicate the difference between 'FP' and 'No FP' with respect to the matching parameters. Detailed quality of matching tests for all sociodemographic indicators used for matching can be found in Tables III-1 in Supplement 3 in S1 File. Histograms for the distribution of propensity scores can be found in Fig III-2, Supplement 2 in S1 File.

utilization of participants with vocational training, influencing numerous other variables, including high school grades and prior exam success.

The columns 'Mean Bias' and 'Median Bias' provide a further descriptive measure of how different the control and treatment groups are in the matching variables. Here too, matching substantially reduces bias, but does not entirely eliminate it. Tables III-1 in Supplement 3 outline in S1 File, for all control variables in each OSCE station or medical skill, how matching adjusts the composition of the FP and No FP groups. In most cases, bias is (significantly) reduced. However, in some instances, imbalances in one or two variables increase in order to minimize overall bias.

A graphical representation is depicted in Fig III-2, showing histograms of the Propensity Score distribution for the FP and No FP groups. For OSCE stations and medical skills where some bias persists even after matching, the distribution is less homogeneous, with an increased number of cases having very low or very high Propensity Scores in the FP group. This signifies instances where suitable matching partners in the control group could not be identified (FP, off support), and thus, these cases were not considered for the ATT.

The selection of variables for matching is guided both by argumentation and by availability, emphasizing observability. Even with high matching quality, there always remains a statistical uncertainty that is challenging to quantify. To address this uncertainty, a sensitivity analysis was conducted to assess to which degree potential unobservable determinants could be distorted before the identified treatment effects become insignificant.

To examine the robustness of the results, Rosenbaum-Bounds [42, 43] were employed, as detailed in Supplement 3 in S1 File. In essence, Rosenbaum-Bounds model a parameter, Γ, which represents the degree of distortion in the assignment of subjects to treatment and control groups due to both observed and unobserved characteristics. A Γ of 1 indicates a randomized sample, with no difference between treatment and control groups and no confounding. The larger Γ, the more substantial the modeled imbalance between treatment and control groups [27]. A treatment effect is considered 'robust' if the null hypothesis (the assumption that the entire effect is due to confounding) is rejected even for larger Γ. Significant treatment effects with a Γ of 2 are described as very robust [44].

As we are estimating positive treatment effects, our focus is primarily on potential overestimation. The upper-bound significance level (sig+) in Tables III-3 in S1 File indicates whether the assumption of overestimation must be rejected. Unlike PSM, Rosenbaum-Bounds are median-based, which may lead to variations in the significance level [27].

The calculations of Rosenbaum-Bounds in Supplement 3 in S1 File reveal three categories for treatment effects in our study: Firstly, for OSCE stations and skills such as Physical examination, PVC, rectal examination, and IM injections, FP has a substantial effect on OSCE performance, even when large confounding is modeled (Table III-3a, III-3g, III-3i, III-3j in S1 File). Secondly, there are OSCE stations, including neurological examination, anamnesis, and diagnosis, where FP has no effect on exam success regardless of the equality or inequality between the treatment and control groups (Tables III-3b-d in S1 File). Thirdly, for the medical skills venipuncture and ECG, treatment effects become significant only when large confounding is modeled. This suggests that the observed differences between FP and No FP are explained by unobserved factors.

We further tested the robustness of our results by varying the matching parameters. For instance, we examined scenarios involving solely sociodemographic factors, including participants' age, or exclusively factors associated with exam success. Additionally, for the OSCE stations Physical examination, neurological examination, Anamnesis, and Diagnosis we tested the division between "practiced only in the Skills Lab" and "FP only outside the Skills Lab" (i.e., the data collected in the year 2023). None of these variations significantly altered the estimated ATT (not shown in tables).

## Discussion

In this study, we empirically tested the concept 'assessment drives learning' in the context of voluntary, free practice (FP) for an undergraduate OSCE in the second year of study at Hannover Medical School. We examined whether the anticipation of a complex assessment motivated students to prepare accordingly, and whether this training had positive effects on performance in the respective parts of the OSCE. To avoid interfering with students' exam preparation and in order to be able to draw causal inferences, we employed the quasi-experimental approach of Propensity Score Matching (PSM).

### Summary of results and interpretation

We showed that the complexity of the OSCE format as well as the importance of the assessed competencies indeed served as strong motivators for MHH students to learn and to apply them, corroborating previous findings by Buss et al. [45].

We found statistically significant and robust, positive treatment effects on exam performance for FP of physical examinations and some (PVC, injections, digital-rectal examination), but not all medical skills. We think that the complexity of a task might explain (besides statistical explanations) the presence or absence of a positive effect of FP. Venipuncture and venous

catheter placement are a good example: while they appear very similar, PVC entails more steps that have to be considered (e.g. fixating the cannula, insertion of the mandrin, clamping the vein).

We also find a robust, positive treatment effect for physical examinations of the heart, lungs, abdomen, etc.; individuals with FP scored almost five percentage points higher in the OSCE section 'Physical examination' compared to individuals from the control group.

We do not find significant differences for the 'Neurological examination'. This is somewhat surprising, as physical and neurological examinations are very similar in terms of procedure and requirements. Both stations presume broad medical knowledge and include symptom-oriented medical histories that exceed the complexity of demonstrating medical skills. However, we argue that our data provide some hints that can explain the differences between internal and neurological examinations: physical examinations were practiced more frequently within the Skills lab than the neurological examination. We argue that a Skills lab as a mock-up clinic offers more realism than private premises. Another explanation is that there are more curriculum-based teaching units for internal examinations than for the neurological examination. This can lead to students being better prepared for the physical examination even without FP, resulting in better exam results. Since FP is done without guidance from lecturers or trained personnel, it may therefore be more difficult to practice content independently and in a structured manner for the neurological examination.

For doctor-patient communication (anamnesis and diagnosis), we found no differences in OSCE performance between the FP and No FP group. However, our results revealed associations between the general willingness to engage with taught contents and performance in the examination. We offer the following explanations for this finding: conversational stations pose greater challenges than medical skills, which can be trained independently, i.e. without assistance, and have clear-cut rules for point allocation. Practice of anamnesis and diagnosis always requires a second person; this person, playing the patient, must also perform the role earnestly, which may not always be the case due to, for example, lack of training–in opposition to simulated patients in the OSCE. Familiarity between 'examiner' and 'examinee' in practice situations may be another hindrance. Although conversational stations have a standardized scope of expectations and list of topics outlined in a checklist, each conversation varies, guided by both the examinee and the standardized patient. Recurring elements such as common phrases, greetings, and general aspects of anamnesis and diagnosis disclosure can be practiced, but doctor-patient interactions remain much more complex and unpredictable compared to medical skills, which usually involve a set of routine maneuvers.

Success at the communication stations likely reflects to a large degree (higher than for other parts of the OSCE) a general diligence and determination to engage with course contents.

## Contextualization

Practical-technical and communicative skills are as fundamental to medical practice as the utilization of theoretical knowledge. Yet, in medical education, practical and interpersonal competencies have traditionally played a rather subordinate role and are even less frequently assessed and evaluated. Previous research with randomized controlled trials (RCT) has shown that targeted practice leads to better and more expertly execution of medical skills [46–48], while other studies found correlations between practice and confidence in ones competencies [3, 20]. However, up to date, there has been very little evidence whether voluntary, extracurricular practice of medical-practical skills has direct, positive effects on exam performance e.g. in an Objective Structured Clinical Examination (OSCE), which can assess the development of professional competencies in a highly standardized form. With our study, we addressed this

research gap by estimating unbiased, robust treatment effects with Propensity Score Matching (PSM).

A major advantage of PSM is that it addresses confounding factors and biases inherent to observational data. In the context of our study design, it was, for example, reasonable to assume that diligent students, who can typically expect good grades in exams, were also more likely to use the Skills lab for preparation. Failing to consider such parameters can lead to a claimed positive effect of practice that is actually due to high GPA or success in prior exams positively influencing OSCE results [18, 49]. In our study, we corroborated previous findings that students with vocational training or professional experience in adjacent healthcare professions have advantages in certain aspects of the OSCE, especially in the demonstration of medical skills, compared to their classmates with an excellent GPA [18], who entered medical school immediately after graduation. These students may not necessarily need to refresh skills such as blood collection or venous catheter placement.

Irrespective of these considerations and our findings that a complex exam like an OSCE does indeed motivate additional, extracurricular practice of crucial skills and techniques and has positive effects on study success, consistent practice is likely imperative for consolidating any learning achievement. In the academic curriculum, refreshment and consolidation occur formally through clerkships and internships. Whether the skills acquired during OSCE preparation are sustained, or partially forgotten by the end of the program, remains unanswered in this study. Observations from the Skills lab at MHH suggest that students, without the motivation of the OSCE, rarely engage in FP in later years of study. Thus, some routines learned in the second year of study may be forgotten when students graduate [50]. The regular performance of more practically oriented assessments would foster the continuation and internalization of practical-technical skills in the sense of 'assessment drives learning' [11]. Although high personnel, time, and financial costs may set limits in this regard [51], a second compulsory OSCE is set to be introduced in 2025 at MHH and will test a wide array of theoretical, practical and communicative skills towards the end of the study program. Whether the second OSCE positively influences students' willingness to continuously engage in voluntary practice, will become a very interesting research endeavor in a few years' time.

## Strengths and limitations

In our study, we estimated statistically robust treatment effects of voluntary practice on OSCE results in the setting of a *real* clinical examination of undergraduate students with observational data. With our Propensity Score Matching (PSM) approach, we were able to reduce potential biases and showed that simple statistical sample tests tended to overestimate true effect sizes. Our analysis further showed that a positive relationship between practice and exam success is not a universally valid automatism. Rather, the presence and strength of effects depended on the type, complexity of and familiarity with the task.

Some limitations of this study must be acknowledged: our sample comprises less than one-third of eligible participants, although it is quite representative of the three included cohorts. In particular for some less-frequently assessed medical skills, the sample size is small, and matching resulted, on some occasions, in the removal of further observations to obtain precise, robust and unbiased estimates. As some parts of the OSCE were practiced by either a large or a very small share of participants, there are some imbalances in the size of control- and treatment groups, which we countered with matching with replacement. Another limitation is that data on practice behavior and Skills lab visits relied on self-declarations and required participants to recall their OSCE preparation from memory. For this reason, we refrained from asking how often each skill was trained, and we could not verify if participants' answers were

always accurate. Although the OSCE at MHH is highly standardized, our results are not fully generalizable, as timing and topics are university-specific, which is also the case for the equipment of the Skills lab.

## Supporting information

**S1 File. Supplementary information: 1) sample statistics; 2) further results; 3) quality of matching and robustness checks; 4) questionnaire.**
(PDF)

**S1 Data.**
(XLSX)

**S1 Checklist. Human participants research checklist.**
(DOCX)

## Acknowledgments

We thank all respondents for their participation in this study. We appreciate the help by Christoph Noll and Sina Golon in the provision of data on OSCE results and information on exam contents and OSCE procedures. We thank two reviewers and participants at the 2022 annual meeting of the GMA (German Association for Medical Education) for helpful comments and suggestions.

## Author Contributions

**Conceptualization:** Stefanos A. Tsikas, Kambiz Afshar, Volkhard Fischer.

**Data curation:** Stefanos A. Tsikas.

**Formal analysis:** Stefanos A. Tsikas.

**Investigation:** Stefanos A. Tsikas.

**Methodology:** Stefanos A. Tsikas.

**Project administration:** Stefanos A. Tsikas.

**Software:** Stefanos A. Tsikas.

**Validation:** Stefanos A. Tsikas, Kambiz Afshar, Volkhard Fischer.

**Visualization:** Stefanos A. Tsikas.

**Writing – original draft:** Stefanos A. Tsikas, Kambiz Afshar.

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
