## [Decision Letter · Decision Letter 0]

24 Jul 2024

PONE-D-24-14286Does voluntary practice improve the outcome of an OSCE in undergraduate medical studies? A Propensity Score Matching approachPLOS ONE

Dear Dr. Tsikas,

Thank you for submitting your manuscript to PLOS ONE. After careful consideration, we feel that it has merit but does not fully meet PLOS ONE’s publication criteria as it currently stands. Therefore, we invite you to submit a revised version of the manuscript that addresses the points raised during the review process.

You have done good research but you. have to revise the manuscript in the line of comments of both reviewers. Thanks and, regards

We look forward to receiving your revised manuscript.

Kind regards,

Rano Mal Piryani, MBBS, MCPS, DTCD, MD, Fellowship in Med Education

Academic Editor

PLOS ONE

Journal Requirements:

2. In the online submission form, you indicated that "Data and code will be made available by the corresponding author upon reasonable request."

**Additional Editor Comments:**

Dear Authors

You have done good research but you. have to revise the manuscript in the line of comments of both reviewers.

Thanks and, regards

Reviewers' comments:

Reviewer's Responses to Questions

**Comments to the Author**

1. Is the manuscript technically sound, and do the data support the conclusions?

Reviewer #1: Partly

Reviewer #2: Partly

2. Has the statistical analysis been performed appropriately and rigorously? 

Reviewer #1: No

Reviewer #2: Yes

3. Have the authors made all data underlying the findings in their manuscript fully available?

Reviewer #1: Yes

Reviewer #2: Yes

4. Is the manuscript presented in an intelligible fashion and written in standard English?

Reviewer #1: Yes

Reviewer #2: Yes

5. Review Comments to the Author

Reviewer #1: "The informed consent and study regulations governing the use of personal data for evaluation/research and quality assurance purposes (in particular sect. 14, para. 1-5 ‘MHH Immatrikulationsordnung’ and sect. 17, para. 3 NHG (Higher Education Act in Lower Saxony, Germany)) rendered a separate approval by an ethics committee unnecessary. The research presented here is in accordance with the Helsinki Declaration"- I do not understand this statement. Even if its exempt, you need to apply to institutional review board committee for approving the exempt status of research globally. If this is not the case in Germany-this is beyond my understanding.

Secondly, even though ANOVA is a test of Association that can be applied when the categorical variable is non-binary. In this manuscript, to my understanding, categorical variables are binary and not non-binary.

Thirdly, introduction can be shortened. Even though the description about the entire OSCE is beautiful, it becomes monotonous to read about OSCE. A greater focus on the overall study variables would shorten the descriptive part of the manuscript.

Fourthly, discussion may be tightened up a little bit.

Reviewer #2: Investigation is a good topic. However, researchers must update the following.

1. The study's justification must be strengthened, and the topic of inquiry must be strengthened by a review of the literature. This study lacks a theoretical or conceptual framework that would enable readers to comprehend the genesis and processing of this notion within the context of the research issue.

2. When it comes to methodology, the researcher needs to explain and provide support for the population, sample, sampling strategy, and sample size computation. Verify the instrument's validation as well, please.

6. PLOS authors have the option to publish the peer review history of their article (what does this mean?). If published, this will include your full peer review and any attached files.

Reviewer #1: **Yes: **Dr. Saira Akhlaq

Reviewer #2: **Yes: **Fozia Fatima (Assistant Professor), Department of Health Professions Education, National University of Medical Sciences, Pakistan

---

## [Author Response · Author response to Decision Letter 0]

27 Aug 2024

Please find the separate document we have uploaded for detailed responses to all reviewer comments.

---

## [Editor Report · Decision Letter 1]

7 Oct 2024

Does voluntary practice improve the outcome of an OSCE in undergraduate medical studies? A Propensity Score Matching approach

PONE-D-24-14286R1

Dear Dr. Tsikas,

We’re pleased to inform you that your manuscript has been judged scientifically suitable for publication and will be formally accepted for publication once it meets all outstanding technical requirements.

Kind regards,

Rano Mal Piryani, MBBS, MCPS, DTCD, MD, Fellowship in Med Education

Academic Editor

PLOS ONE

Additional Editor Comments (optional):

Thanks for addressing the comments of reviewers.
---

## [Editor Report · Acceptance letter]

14 Oct 2024

PONE-D-24-14286R1 

PLOS ONE

Dear Dr. Tsikas, 

I'm pleased to inform you that your manuscript has been deemed suitable for publication in PLOS ONE. Congratulations! Your manuscript is now being handed over to our production team.

Kind regards, 

on behalf of

Dr. Rano Mal Piryani 

Academic Editor

PLOS ONE